



# Characterisation of African biomass burning plumes and impacts on the atmospheric composition over the South-West Indian Ocean

Bert Verreyken[1,2,3], Crist Amelynck[1,2], Jérôme Brioude[3], Jean-François Müller[1], Niels Schoon[1], Nicolas Kumps[1], Aurélie Colomb[4], Jean-Marc Metzger[5], Christopher F. Lee[6,7], Theodore K. Koenig[6,7], Rainer Volkamer[6,7], and Trissevgeni Stavrakou[1]

[1]Royal Belgian Institute for Space Aeronomy, B-1180 Brussels, Belgium
[2]Department of Chemistry, Ghent University, B-9000 Ghent, Belgium
[3]Laboratoire de l'Atmosphère et des Cyclones, UMR 8105, CNRS, Université de La Réunion, 97744 Saint-Denis, France
[4]Laboratoire de Météorologie Physique, UMR6016, CNRS, Université Clermont Auvergne, 63178 Aubière, France
[5]Observatoire des Science de l'Univers de La Réunion, UMS3365, 97744 Saint-Denis, France
[6]Cooperative Institute for Research in Environmental Sciences (CIRES), University of Colorado, Boulder, CO, USA
[7]Department of Chemistry, University of Colorado, Boulder, CO, USA

**Correspondence:** B. Verreyken (bert.verreyken@aeronomie.be)

**Abstract.** We present an investigation of biomass burning (BB) plumes originating from Africa and Madagascar based on measurements of carbon monoxide (CO), ozone ($O_3$), nitrogen dioxide ($NO_2$) and a suite of volatile organic compounds (VOCs) obtained during the dry season of 2018 and 2019 at the high altitude Maïdo observatory ($21.1°$ S, $55.4°$ E, 2250 m above sea level), located on the remote island of La Réunion in the South-West Indian Ocean (SWIO). Biomass burning

plume episodes were identified from increased acetonitrile ($CH_3CN$) mixing ratios. Enhancement ratios (EnRs) — relative to CO — were calculated from in situ measurements for $CH_3CN$, acetone ($CH_3COCH_3$), formic acid (HCOOH), acetic acid ($CH_3COOH$), benzene ($C_6H_6$), methanol ($CH_3OH$) and $O_3$. We compared the EnRs to emission ratios (ERs) — relative to CO — reported in literature in order to estimate loss/production of these compounds during transport. For $CH_3CN$ and $CH_3COOH$, the calculated EnRs are similar to the ERs. For $C_6H_6$ and $CH_3OH$, the EnR is lower than the ER, indicating a significant

net sinks of these compounds. For $CH_3COCH_3$ and HCOOH, the calculated EnRs are larger than the ERs. The discrepancy reaches an order of magnitude for HCOOH ($18 – 34$ pptv ppbv$^{-1}$ compared to $1.8 – 4.5$ pptv ppbv$^{-1}$). This points to significant secondary production of HCOOH during transport. The Copernicus Atmospheric Monitoring Service (CAMS) global model simulations reproduces well the temporal variation of CO mixing ratios at the observatory but underestimates $O_3$ and $NO_2$ mixing ratios in the plumes on average by 16 ppbv and 60 pptv respectively. This discrepancy between modelled and measured

$O_3$ mixing ratios was attributed to i) large uncertainties in VOC and $NO_x$ ($NO+NO_2$) emissions due to BB in CAMS and ii) misrepresentation of $NO_x$ recycling in the model during transport. Finally, transport of pyrogenically emitted CO is calculated with FLEXPART in order to i) determine the mean plume age during the intrusions at the observatory and ii) estimate the impact of BB on the pristine marine boundary layer (MBL). By multiplying the excess CO in the MBL with inferred EnRs at the observatory, we calculated the expected impact of BB on $CH_3CN$, $CH_3COCH_3$, $CH_3OH$ and $C_6H_6$ concentrations in

the MBL. These excesses constitute increases of $\sim 20\% – 150\%$ compared to background measurements in the SWIO MBL reported in literature.





## 1 Introduction

Non-methane volatile organic compounds (NMVOCs) are key tropospheric constituents. Many of them are highly reactive with the major atmospheric oxidants, especially with the OH radical, and therefore they strongly affect the oxidation capacity of the troposphere (Atkinson, 2000). By being a strong sink for OH, they also exert control on the lifetime of methane (Zhao et al., 2019) and thus on climate. Moreover, OH-initiated NMVOC oxidation modulates tropospheric $O_3$ concentrations and is the major source of this secondary pollutant in high $NO_x$ (NO+$NO_2$) environments (Monks et al., 2015). Less volatile NMVOC oxidation products contribute to the formation and growth of secondary organic aerosol which deteriorates air quality and affects radiative forcing, and hence climate, both in a direct (by interacting with solar radiation) and indirect way (by acting as cloud condensation nuclei) (IPCC, 2013).

Whereas atmospheric oxidation of precursor VOC species is the dominant source of many oxygenated VOCs (OVOCs), primary anthropogenic emissions and (bidirectional) exchange with the biosphere and the ocean and biomass and biofuel burning also contribute to the atmospheric OVOC burden (Mellouki et al., 2015). Photochemical degradation and dry and wet deposition are the major sink processes. Global OVOC budgets are still prone to large uncertainties due to an incomplete understanding of photochemical production and loss processes and ocean–atmosphere exchange (Millet et al., 2010; Fischer et al., 2012; Read et al., 2012; Wang et al., 2019), and a paucity of (O)VOC data, especially at remote marine areas where the oxidative capacity of the atmosphere is mainly controlled by OVOCs (Lewis et al., 2005; Carpenter and Nightingale, 2015).

The South-West Indian Ocean (SWIO) is one of the few pristine regions on Earth. It is largely decoupled from emissions originating from large bodies of land and is well suited to characterise remote marine air composition and ocean emissions (Colomb et al., 2009; Mallet et al., 2018). Located in the SWIO is the French overseas department La Réunion, a small volcanic island, home to the high altitude Maïdo atmospheric observatory (21.1° S, 55.4° E, 2250 m above sea level) (Baray et al., 2013), hereafter referred to as RUN. From October 2017 to November 2019, a high-sensitivity quadrupole-based Proton Transfer Reaction Mass Spectrometry VOC analyser (hs-PTR-MS) was deployed at RUN in the framework of the OCTAVE (Oxygenated Compounds in the Tropical Atmosphere: Variability and Exchanges) project (http://octave.aeronomie.be). In combination with other ground-based and satellite data, the resulting near-continuous high time-resolution two-year data set will serve to better constrain VOC emissions in the remote tropical marine atmosphere and to identify missing sources. Part of this dataset has already been used in a source apportionment study of formaldehyde (HCHO) (Rocco et al., 2020).

The present paper contributes to the disentanglement of the different sources contributing to the (O)VOC composition at RUN by focusing on the role of biomass burning (BB). It is established from ground-based remote-sensing Fourier Transform Infrared (FTIR) observations that BB impacts the atmosphere over La Réunion. The BB events affecting the region occur most frequently in southern Africa and Madagascar but impacts from burning in South America and Malaysia has also been identified (Duflot et al., 2010; Vigouroux et al., 2012). Seasonality of in situ CO concentrations at RUN indicates that BB plumes also impact the atmospheric composition at the surface (Zhou et al., 2018). This was confirmed by the hs-PTR-MS dataset generated at RUN for the OCTAVE project (Fig. A1). Biomass burning represents the second largest global source of NMVOC emissions (Yokelson et al., 2008; Akagi et al., 2011). Pyrogenic emissions are reasonably well constrained by numerous lab-





oratory studies (e.g. Holzinger et al. (1999); Christian et al. (2003); Yokelson et al. (2008)) and observations of BB plumes in the atmosphere (e.g. Lefer et al. (1994); Yokelson et al. (1999, 2003); Jost et al. (2003); de Gouw et al. (2006); Vigouroux et al. (2012); Akagi et al. (2014)). Emission factors for numerous compounds have been compiled recently by Andreae (2019). The investigation of compositional changes during BB plume transport may provide valuable clues for identifying missing sources

of reactive trace species (e.g. Jost et al. (2003); de Gouw et al. (2006); Chaliyakunnel et al. (2016)). This is of particular interest for the carboxylic acids, as current models underestimate their observed abundances, possibly in part due to a misrepresentation of the contribution of biomass burning (Paulot et al., 2011; Chaliyakunnel et al., 2016).

In this work, we focused on the first BB plume intrusions from the fire season in 2018 and 2019. Enhancement ratios (EnRs) — relative to excess CO — of excess acetonitrile ($CH_3CN$), formic acid ($HCOOH$), acetone ($CH_3COCH_3$), acetic acid

($CH_3COOH$), benzene ($C_6H_6$), methanol ($CH_3OH$) and $O_3$ were calculated from in situ measurements at RUN for the first time. In addition, observations of BB plumes at RUN were used to evaluate the global near-real time (NRT) CO, $O_3$ and $NO_2$ modelled concentrations at RUN from the Copernicus Atmospheric Monitoring Service (CAMS). Finally, we propose a way to use in situ VOC measurements at RUN to estimate the impact of BB plumes on the pristine marine boundary layer (MBL) over the SWIO. This is done for $CH_3CN$, $CH_3COCH_3$, $C_6H_6$ and $CH_3OH$.

In section 2 the instruments, methods and models used in this study are presented. The results are shown in section 3 and discussed in section 4.

## 2   Observations and Methods

### 2.1   Observations

#### 2.1.1   In situ air mass characterisation

RUN houses routine instruments characterising in situ air constituents in the context of global networks such as GAW (Global Atmospheric Watch), ICOS (Integrated Carbon Observation System) and ACTRIS (European Research Infrastructure for the observation of Aerosol, Clouds and Trace Gases). A summary of routine measurements from the observatory used in this study is shown in Table 1. A detailed description of these and other operational routine instruments at the observatory can be found in Duflot et al. (2019); Baray et al. (2013) and Zhou et al. (2018).

In the frame of the OCTAVE project, a hs-PTR-MS instrument (Ionicon Analytik GmbH, Austria) was deployed at RUN from October 2017 to November 2019. This resulted in a near-continuous high time-resolution two-year data set of volatile organic compounds (VOCs). The instrument was run in the multiple ion detection mode using $H_3O^+$ precursor ions with a total cycle time of $\sim$ 2.7 min. Regular calibrations of the hs-PTR-MS were performed by diluting a gravimetrically prepared VOC/$N_2$ mixture (Apel-Riemer Environmental Inc., Miami, FL, USA; stated accuracy of 5% for all VOCs) with zero-VOC air

obtained by sending ambient air through a catalytic converter (Parker, type HPZA-3500, Haverhill, MA, USA). This resulted in VOC concentrations in the lower ppbv range. Calibrations as a function of relative humidity were performed bimonthly by controlling the humidity of the zero air with a dew point generator (LI-COR LI610, Lincoln, Nebraska, USA). The calibration





factor (CF) for acetic acid (CH$_3$COOH) was estimated from the experimentally determined CF for CH$_3$COCH$_3$. This is done by considering the calculated collision rate constants of H$_3$O$^+$ with CH$_3$COOH and CH$_3$COCH$_3$ (Su, 1994), the contributions

of the protonated molecules to the respective product ion distributions (Schwarz et al., 2009; Inomata and Tanimoto, 2010), and by assuming the same hs-PTR-MS transmission efficiency for ions with a mass difference of 2 u. Similarly, the CF of HCOOH was determined from the measured one of acetaldehyde. The humidity dependence of formic and acetic acid CFs obtained at similar hs-PTR-MS operating conditions has been reported in literature (Baasandorj et al., 2015) and has been taken into account for quantification. By considering the uncertainties on the different parameters involved in the carboxylic

acid quantification in the present study, the total uncertainty on their mixing ratio is estimated at 50%. Measurements were averaged over 1 hour to lower the limit of detection (LoD) and the random fluctuations of the measurements. A list of masses, and their associated compound(s), recorded by the hs-PTR-MS together with the LoD, dwell time and whether the compounds are directly calibrated is shown in Table 2.

### 2.1.2 Ground-based remote sensing

The University of Colorado Multi-AXis Differential Optical Absorption Spectroscopy (CU MAX-DOAS) instrument consists of a scanning (horizon – zenith – horizon) telescope coupled to two ultraviolet-visible grating spectrometers (Coburn et al., 2011). Scattered-light solar spectra are collected along lines of sight at different elevation angles above the horizon (Hönninger et al., 2004), and analyzed using DOAS least-square fitting (Platt and Stutz, 2008) to retrieve trace gas slant column densities (SCD) by the QDOAS software package (Danckaert et al., (accessed June 10, 2019). For this analysis, NO$_2$ (Vandaele et al.,

1998) and O$_2$–O$_2$ (Thalman and Volkamer, 2013) were retrieved in a fitting window from 425 – 490 nm, using the further fit settings as described in Kreher et al. (2020). Near-surface volume mixing ratios of NO$_2$ were retrieved from limb (0° elevation angle) spectra following Dix et al. (2016). This approach takes advantage of the fact that the limb viewing geometry is highly sensitive to absorbers near instrument altitude. O$_2$–O$_2$ is used to parameterise aerosol extinction near instrument altitude, avoiding the need for complex aerosol profile information (Sinreich et al., 2013; Dix et al., 2016). The NO$_2$ profile shape

was constructed using a typical tropical background with BB enhancements collocated to excess CO from FLEXPART (see section 2.3.2). Variations on the retrieval settings and profile assumptions indicate that ∼10 pptv NO$_2$ can be quantified with an uncertainty of 5 pptv using this approach. Further tests using NO$_2$ and O$_4$ fits at shorter wavelengths (Kreher et al., 2020) determined that the retrieved NO$_2$ volume mixing ratios generally agree within the reported uncertainty, despite the different spectral ranges average NO$_2$ over different horizontal spatial scales. This indicates that the NO$_2$ mixing ratio is representative

of the regional lower troposphere predicted by the CAMS model.

### 2.2 Enhancement ratios

The impact of BB events on an atmospheric species X is often quantified by an emission factor (EF$_X$) or an enhancement ratio relative to a compound Y (EnR$_{X/Y}$). The first is defined as the mass of compound X that is released by burning 1 kg of dry fuel, whereas the second is defined as the excess mixing ratio — due to BB — of compound X ($\Delta$X), with respect to that of

a reference species Y ($\Delta$Y). If the EnR is measured close to the source and/or if both X and Y were minimally affected by


physico–chemical interactions, it is also referred to as the emission ratio of compound X normalised to Y ($ER_{X/Y}$). The ER can be computed from the EF by taking the molecular weights (MW) of both species into account:

$$ER_{X/Y} = \frac{EF_X}{EF_Y} \frac{MW_Y}{MW_X}. \tag{1}$$

A list of EFs with the associated fuel type has been compiled most recently by Andreae (2019). When comparing the EnR

values derived from our observations to ERs from literature, production/loss of plume constituents during transport should be taken into consideration. Enhancement and emissions ratios are often used with CO as the reference species Y. Hereafter, enhancement ratios are always considered with respect to CO unless specifically stated otherwise.

Excess mixing ratios are determined above the background — unaffected by BB — diel profiles which were approximated by the seasonal median diel profiles (appendix A2). During the day, mesoscale transport at La Réunion results in the observatory

being located in the planetary boundary layer (PBL). The chemical composition of the PBL is determined by marine, biogenic and anthropogenic sources and sinks interacting in physicochemical atmospheric processes. At night, air masses arriving at RUN originate primarily from the free troposphere (FT). This mesoscale transport results in a natural diel variation of compound mixing ratios which needs to be taken into account when calculating EnR.

## 2.3 Modelling


Below we discuss the model simulations used in this study. Each model is used with a specific goal in mind. First, we evaluate the CAMS NRT atmospheric composition service using in situ measurements. It is important that CAMS correctly reproduces CO concentrations at RUN as pyrogenic emissions used in this service will be used to calculate transport of excess CO ($\Delta$CO) over the SWIO with the Lagrangian FLEXible PARTicle dispersion model, FLEXPART (Stohl et al., 1998; Stohl and Thomson,

1999; Stohl et al., 2005; Pisso et al., 2019). We use FLEXPART to calculate the mean plume ages during the BB episodes at RUN but also to calculate the impact of pyrogenic emissions on the pristine MBL over the SWIO. Finally FLEXPART-AROME (Verreyken et al., 2019) is used to simulate mesoscale transport in complex terrain towards the observatory. This last simulation is performed in an effort to quantify the PBL–FT mixing during BB intrusions and identify the main transport mode of the plumes.

### 2.3.1 CAMS NRT


The CAMS NRT service was developed based on a series of Monitoring Atmospheric Composition and Climate (MACC) research projects. It provides daily forecasts of reactive trace gases, greenhouse gases and aerosol concentrations. The data are generated by the Integrated Forecast System (IFS) at the European Centre for Medium-Range Weather Forecasts (ECMWF). The chemical mechanism used is an extended version of the Carbon Bond 2005 lumped chemistry scheme (Flemming et al.,

2015). BB emissions implemented in the NRT service rely on the Global Fire Assimilation System v1.2 (GFAS v1.2) inventory. The GFAS assimilates fire radiative power observations from the NASA MODIS satellites to quantify BB emissions (Giuseppe et al., 2018; Rémy et al., 2017; Kaiser et al., 2012). On 9 July 2019, the model was updated to use the CAMS emission



inventories, CAMS_GLOB_ANT v2.1 and CAMS_GLOB_BIO v1.1 (Granier et al., 2019), instead of the previous MACCity (Lamarque et al., 2010) and the MEGAN_MACC (Sindelarova et al., 2014) inventories. BB plume injection heights were also
introduced in this update. A full description and validation of the update was reported by Basart et al. (2019).

We used modelled mass mixing ratios at the location of RUN calculated on different pressure levels (1000, 950, 925, 900, 850, 800, 700 and 600 mbar levels) every three hours (0, 3, 6, 9, 12, 15, 18, 21 UT) from the midnight forecast at $0.5° \times 0.5°$ resolution [1]. The CO, $O_3$ and $NO_2$ mass mixing ratios are transformed to volume mixing ratios and compared to the in situ measurements.

### 2.3.2 FLEXPART

FLEXPART, driven by ECMWF IFS meteorology at $0.5° \times 0.5°$ horizontal resolution was used to calculate the transport of $\Delta CO$ due to BB during 15 June – 31 August 2018 and 17 June – 22 August 2019. The CO emissions are provided by the GFAS v1.2 inventory. Three-hourly mean mixing ratios of CO were generated on vertical layers of 500 m depth between 0 and 3500 m above ground level (a.g.l.). The output was given on a $0.5° \times 0.5°$ grid. Due to the low horizontal resolution, the orographic
profile of La Réunion is not well resolved. For example, the ground level of RUN is only 284 m above sea level (a.s.l.) in the model, much below the true altitude of 2250 m a.s.l.

Age classes (AC) are used to estimate the mean plume age ($T$) for the different intrusions. The CO plumes are categorised by age with 2 day resolution ($T_{AC} = 1 \pm 1, 3 \pm 1, .., 23 \pm 1$ days). BB plume excesses are traced for 24 days, after which the plume is assumed to be diluted to negligible background levels. The mean BB plume age is obtained from the FLEXPART output by:


$$T = \frac{\sum\limits_{j=0}^{11} \Delta CO_j \times T_j}{\sum\limits_{j=0}^{11} \Delta CO_j}, \tag{2}$$

where $\Delta CO_j$ is the mean mixing ratio calculated by FLEXPART with AC=$j$.

To estimate the impact of BB on the MBL for compound X, we use:

$$\Delta X_{estimate} = \Delta CO \times EnR_X, \tag{3}$$

where $\Delta CO$ is calculated by FLEXPART and $EnR_X$ is inferred from data. In this approach, the role of an ocean sink is neglected.

### 2.3.3 FLEXPART-AROME

FLEXPART-AROME 24-hour backtrajectory simulations are used to estimate the respective contribution of the PBL and the free troposphere to the in situ measurements at RUN. Lesouëf et al. (2011) characterised the PBL impact on the Maïdo mountain
region by using a passive boundary layer tracer initialised in an approximation of the minimal boundary layer. This PBL proxy

---

[1] available at https://apps.ecmwf.int/datasets/cams-nrealtime/levtype=pl/





is defined as 500 m a.g.l., capped at 1000 m a.s.l. Here, the inverse approach is used by calculating the fraction of time air parcels have spent in the PBL-proxy during the 24-hour backtrajectory simulation. This fraction measures the potential impact of surface emissions on the in situ measurements. We will split this fraction up according to surface type (land/ocean) and call the separate components the mixing fraction (MF). Given the lack of a high-resolution anthropogenic emission inventory over

La Réunion, we are not able to use the model to quantify mixing ratios unperturbed by BB plumes and instead use the median diel profile as stated in section 2.2.

## 3    Results

### 3.1    Data analysis

Six episodes of enhanced $CH_3CN$, which is a typical BB compound, were identified in August 2018 and August 2019 (Fig. 1).

The correlation ($r$) between the excess mixing ratio of the monitored trace gases and $\Delta CH_3CN$, during the identified intrusions, is shown in Table 3. As dimethyl sulphide (DMS) is only marginally present in pyrogenic emissions (0.0022 – 0.05 g emitted per kg dry matter burned from tropical forest and agricultural residue burning respectively (Andreae, 2019)) and has a short atmospheric lifetime (less than 1 day (Blake et al., 1999)), the correlation between $\Delta DMS$ and $\Delta CH_3CN$ is not expected to be directly related to the BB emissions. For this reason, compounds that correlated less well with $\Delta CH_3CN$ than $\Delta DMS$ were

not considered as plume constituents. Plume constituents in this analysis are thus limited to $CH_3CN$, $HCOOH$, $CH_3COCH_3$, $CH_3COOH$, $O_3$, $C_6H_6$ and $CH_3OH$.

Mean background (i.e. outside BB episodes) concentrations of plume constituents in austral winter together with the mean excesses during the different BB intrusions (in %) are shown in Table 4. Correlation with $CH_3CN$ is especially strong for compounds showing large excesses compared to the diel background pattern (illustrated in appendix A2). We note that trace

species such as HCHO, acetaldehyde ($CH_3CHO$) and methyl ethyl ketone (MEK) show elevated concentrations during the night in BB episodes, which suggests that they are related to BB. However, as the diel patterns for these compounds are subject to strong variability, excesses are poorly characterised during the day and not analysed further here.

For each of the intrusions, the EnR is computed for $CH_3CN$, $CH_3OH$, $CH_3COCH_3$, $C_6H_6$, $HCOOH$, $CH_3COOH$ and $O_3$. Figure 1 shows the scatter plots correlating the excess of the trace species monitored by the hs-PTR-MS instrument and $\Delta CO$.

The calculated EnRs are found in Table 5.

### 3.2    Comparison with model

#### 3.2.1    FLEXPART-AROME

Figure 2 shows the fraction of time spent in the PBL-proxy from Lesouëf et al. (2011) over sea (blue) and land (brown), during

the 24-hour backtrajectory calculations with FLEXPART-AROME, together with the relative humidity (RH) at the observatory. Biomass burning intrusions have lower than average RH values. The humidity peaks during the BB episodes are coincident





with peak impacts of the MBL. It is also shown that the impact of mesoscale PBL emissions on the VOC concentrations is lower during the BB intrusions in August 2018 than in August 2019.

### 3.2.2 CAMS near-real-time model simulations

The modelled mixing ratios at RUN calculated by the CAMS NRT service are compared to data recorded at the observatory for CO, O$_3$ and NO$_2$ (Fig. 3). The model bias for CO, during the BB intrusions, is lowest on the 800 mbar pressure level (bias of 9.7 ppbv), which is closest to the mean pressure measured at the observatory during the same period (792.8 mbar). Note that CAMS reflects well the CO mixing ratios at Maïdo both during and outside (5.1 ppbv bias) BB episodes. As CO is a chemically stable compound in the atmosphere, the agreement between model and measurements indicates that synoptic scale transport

and mesoscale mixing with the BB plumes at the location of RUN is sufficiently reproduced by the CAMS NRT model.
The O$_3$ model bias is 16 ppbv during the BB episodes with a maximum bias of 39 ppbv (67% above the calculated value). Outside the BB episodes, the CAMS O$_3$ concentrations show only a small bias (0.8 ppbv), within the uncertainty of measurements. This good agreement outside of the BB events suggests that mesoscale O$_3$ sources and sinks either have a limited impact or are correctly calculated by the model at the location of RUN.

The NO$_2$ bias reaches 60 pptv during BB episodes, while it is only 9 pptv (within 10 pptv DOAS accuracy error) in other periods. Note that the NO$_2$ measurements are from the ground-based remote sensing CU MAX-DOAS instrument and reflect the NO$_2$ mixing ratio in the lower free troposphere. The large discrepancy in modelled and measured NO$_2$ on 3 August 2019 may be due to a weak BB plume passing near RUN (appendix B).

### 3.2.3 FLEXPART forward simulation

A comparison between $\Delta$CO obtained from measurements and the calculated $\Delta$CO from transport of GFAS v1.2 emission inventory, simulated by FLEXPART, is shown in Fig. 4. Due the misrepresentation of the orographic profile of La Réunion, ground level at the location of the observatory is only 284 m above sea level in the model. The real altitude of RUN (2250 m a.s.l.) is situated near the boundary between layers 1500 – 2000 m a.g.l. and 2000 – 2500 m a.g.l. in the FLEXPART output. In reality, mesoscale transport, not resolved in FLEXPART, mixes the different vertical layers and data recorded at Maïdo

correspond to a mixture between different output levels. In what follows, we consider RUN to be located in the layer between 2000 and 2500 m a.g.l.
The model overestimates $\Delta$CO mixing ratios at RUN by 37 ppbv and 17 ppbv on average during the BB episodes in 2018 and 2019 respectively. Peak differences between modelled and observed mixing ratios are 340 ppbv during the BB episodes in 2018 and 162 ppbv during those in 2019. The model bias outside BB episodes, reduces to 3 ppbv for both 2018 and 2019.

As the timing of BB intrusions is well represented in the model, as can be visually confirmed from Fig. 4, calculated mean plume ages during the different episodes are expected to be accurate. The calculated plume ages are, in chronological order of arrival at RUN, 7.5, 10.6 and 11.3 days in 2018 and 7.4, 9.3 and 13.7 days in 2019.



## 4 Discussion

### 4.1 Transport and dominant sink

The relative humidity during the BB intrusions was generally low (see Fig. 2). Peak RH values correspond to large impact of the MBL and often lower $\Delta CH_3CN$ concentration (e.g. 7 and 17 August 2019, Fig. 2). From this, we expect the plume to be primarily located in the free troposphere, which is drier than the PBL. This is consistent with results from FLEXPART (Fig. 4), where $\Delta CO$ is especially significant in layers above 1500 m a.g.l. The primary sink in the FT during austral winter (dry season) is due to photochemical interaction. This is also evident from the CAMS NRT model where elevated CO mixing ratios are calculated between the 850 mbar and 700 mbar pressure levels ($\sim 1500 - 3000$ m a.s.l.).

### 4.2 Plume characterisation

The emission ratios are computed based on emission factors from (Andreae, 2019) for $CH_3CN$, $HCOOH$, $CH_3COOH$, $CH_3COCH_3$, $C_6H_6$ and $CH_3OH$ are shown in Table 6. Possible fuel types for BB plumes arriving at RUN are: savanna and grassland, tropical forest or agricultural residue. Enhancement ratios are compared to the emission ratios to check for consistency with accepted knowledge regarding sources/sinks during transport.

#### 4.2.1 Acetonitrile, acetone, methanol and benzene

During the synoptic scale transport in the free troposphere, the photochemical sink is expected to be dominant over wet scavenging. As the lifetime with regards to this sink is larger than the maximum plume age (13.7 days) for both $CH_3CN$ ($\tau_{CH_3CN}$= 1.4 years (de Gouw et al., 2003)) and $CH_3COCH_3$ ($\tau_{CH_3COCH_3}$= $36 - 39$ days (Arnold et al., 2005; Fischer et al., 2012)), the EnRs are expected to correspond well with the ERs from literature. This is the case for $CH_3CN$ (Table 6). In contrast, the EnR of acetone ($\sim 8$ pptv ppbv$^{-1}$) is at least a factor of $\sim 2$ larger than the ER from the literature (Table 6), a likely indication of secondary $CH_3COCH_3$ formation in the BB plume. Acetone production has been recorded in BB plumes over the Eastern Mediterranean (Holzinger et al., 2005) and over Namibia (Jost et al., 2003). In contrast, aged BB plumes over Eastern Canada and Alaska did not show evidence of acetone production (de Gouw et al., 2006). Known pyrogenic $CH_3COCH_3$ precursors are propane, i-butane and i-butene (Singh et al., 1994). Using the EFs from Andreae (2019), we find $ER_{propane}$= $1.2 - 3.2$ pptv ppbv$^{-1}$, $ER_{i-butane}$= $0.05 - 0.1$ pptv ppbv$^{-1}$ and $ER_{i-butene}$= $0.30 - 0.52$ pptv ppbv$^{-1}$. Taking these known precursors of secondary $CH_3COCH_3$ into account, as well as acetone formation yields at high $NO_x$ estimated based on the Master Chemical Mechanism MCMv3 (http://mcm.leeds.ac.uk/MCM/) (Saunders et al., 2003), the secondary production of acetone can be estimated. It is found to enhance the acetone EnR by $1.16 - 2.80$ pptv ppbv$^{-1}$, therefore explaining the major part of the discrepancy. This is at odds with results from Jost et al. (2003) where fast $CH_3COCH_3$ production is observed and propane could not be considered as a precursor since this conversion is a slow process.

Both methanol and benzene have shorter expected lifetimes compared to the age of the BB plume arriving at RUN ($\tau_{CH_3OH}$=





7 days (Jacob et al., 2005), $\tau_{C_6H_6}$ = 9 days (Monod et al., 2001)). This is consistent with the reduced EnRs inferred from data at RUN compared to the reported average emission ratios from literature (Table 6).

### 4.2.2 Carboxylic acids

Due to the relatively short global average atmospheric lifetime of HCOOH ($\tau_{\mathrm{HCOOH}}$= 2 − 4 days (Stavrakou et al., 2012)) and
CH$_3$COOH ($\tau_{\mathrm{CH_3COOH}} \approx$ 2 days (Khan et al., 2018)), EnRs in aged BB plumes should not be compared to emissions ratios from literature (Paulot et al., 2011). However, as wet- and dry deposition are dominant sinks for both CH$_3$COOH and HCOOH, their effective lifetime during transport in the FT is expected to be much longer ($\tau_{\mathrm{HCOOH}} \approx$ 25 days from photochemical oxidation (Millet et al., 2015)).

The much higher HCOOH enhancement ratio estimated from RUN data (20 − 30 pptv ppbv$^{-1}$) compared to reported emis-
sion ratios (2 − 4 pptv ppbv$^{-1}$) points to significant secondary production during transport to RUN. A potential precursor to HCOOH strongly emitted by agricultural residue burning is glycolaldehyde (ER = 19 ± 12 pptv ppbv$^{-1}$ (Andreae, 2019)). The yield of HCOOH from glycolaldehyde oxidation has been measured to be 18% at 296 K and 52% at 233 K (Butkovskaya et al., 2006). This may account for part of the HCOOH production during transport. However, recent studies indicate that this production is effective only in high NO$_x$ conditions that are not realistic in a natural environment (Orlando et al., 2012;
Orlando and Tyndall, 2020). Production of HCOOH from glycolaldehyde is thus most likely only a minor source. No other known precursors were identified to account for the high HCOOH production during transport to RUN suggesting a missing source in current knowledge.

Secondary production of HCOOH was also found in BB plumes over Canada (Lefer et al., 1994) but was not observed in previous ground-based FTIR studies at La Réunion (Vigouroux et al., 2012). Enhancement ratios of HCOOH calculated from
the Tropospheric Emission Spectrometer instrument aboard the NASA's aura spacecraft over Africa ranged from 26 to 28 pptv ppbv$^{-1}$ (Chaliyakunnel et al., 2016), consistent with our results. This secondary HCOOH production in BB plumes could account for part of the discrepancy in global HCOOH budget between models and observations (Chaliyakunnel et al., 2016). As these EnRs are inferred from data over biomass burning hotspots in Africa, HCOOH is probably formed primarily close to the source and conserved during synoptic scale transport towards RUN.
For CH$_3$COOH the enhancement ratio (EnR$_{\mathrm{CH_3COOH}} \approx$14 pptv ppbv$^{-1}$) is of the same order of magnitude as the emission ratios from literature (Table 6). Therefore, in contrast with the case of HCOOH, no significant secondary production of acetic acid in BB plumes is identified.

### 4.2.3 Ozone and NO$_2$

It is generally accepted that O$_3$ is produced in BB plumes during transport (Taupin et al., 2002; Jaffe and Wigder, 2012; Parrington et al., 2013; Arnold et al., 2015; Brocchi et al., 2018). The EnRs obtained in this study (410 − 640 pptv ppbv$^{-1}$) are in agreement with the range of EnRs obtained in tropical BB plumes older than 5 days, compiled by Jaffe and Wigder (2012),





410 – 750 pptv ppbv$^{-1}$.

Figure 3 shows that the CAMS model reproduces correctly the $O_3$ concentrations at RUN outside the BB episodes but under-
estimates $O_3$ during these episodes. The large underestimation of $O_3$ during these episodes indicates a misrepresentation of the
BB emissions at the source and/or missing $O_3$ production during transport in the chemically complex plumes.

The $O_3$ production in the troposphere is highly dependent on the ratio between VOCs and $NO_x$. The CAMS NRT service is
known to overestimate $NO_2$ over southern Africa in austral winter/spring (Flemming et al., 2015; Basart et al., 2020). How-
ever, this overestimation was reduced since the upgrade in 2017 (Basart et al., 2020). Total BB VOC emissions in the IFS of
ECMWF was $\sim 40$ Tg in the year 2008 (Flemming et al., 2015). This is too low in comparison with the top-down estimate by
Stavrakou et al. (2015) where the global pyrogenic VOC emissions are estimated to be $67 - 75$ Tg yr$^{-1}$.

Ozone production in BB plumes tends to be $NO_x$-limited (Jaffe and Wigder, 2012). The measured $NO_2$ mixing ratio during BB
episodes is significantly higher than those calculated by CAMS (Fig. 3). The largest and smallest difference between model and
measurements for both $NO_2$ and $O_3$ were recorded during the first and last BB intrusion in 2019 respectively. This mismatch
for $NO_2$ may be caused by an underestimation of $NO_x$ emissions by fires or by a misrepresentation of $NO_x$ recycling (e.g.
through peroxyacetyl nitrate or PAN). BB plumes reaching RUN are located at relatively low altitudes where warmer tempera-
tures make thermal decomposition of PAN a likely source of $NO_x$. This could be a decisive factor in harmonising modelled and
recorded $O_3$ mixing ratios as an increase in VOC emissions related to BB is unlikely to lead to $O_3$ production in the absence
of $NO_x$.

Uncertainties on VOC and $NO_x$ emissions by BB and misrepresentations of $NO_x$ recycling during transport are both likely
contributors to the misrepresentation of $O_3$ mixing ratios at the location of RUN.

### 4.3    Plume dispersion over the SWIO

Transport of BB plumes recorded by the hs-PTR-MS at RUN takes place primarily in the lower FT. This implies that disper-
sion of the plume into the MBL is possible through turbulent mixing in shallow cumulus clouds and development of the MBL.
Figure 5 shows $\Delta CO$ due to pyrogenic emissions from plumes between 4 and 16 days old (corresponding to the extremes of
plume ages observed at Maïdo) as calculated with FLEXPART on the model output layer $0 - 500$ m a.g.l. By using equation 3,
estimates of $\Delta CH_3CN$, $\Delta CH_3COCH_3$, $\Delta CH_3OH$ and $\Delta C_6H_6$ in the pristine marine boundary layer environment were made
(Fig. 5). To illustrate the importance of these BB plumes on the MBL composition, these expected excesses are compared with
background VOC measurements performed in the SWIO during the MANCHOT campaign that took place December 2004
(Colomb et al., 2009). Shipborne measurements of VOC concentrations were performed South of La Réunion to characterise
the impact of oceanic fronts on MBL composition (Colomb et al., 2009). We use background measurements North (zone I,
$24.2° - 30.2°$ S) and South (zone III, $45.9° - 49.2°$ S) of the different oceanic fronts that were under consideration (Colomb
et al., 2009). Due to the higher concentrations of anthropogenic tracers in zone I of the campaign, it was suggested that there
may have been an impact of African outflow on these backgrounds (Colomb et al., 2009). Note that MANCHOT took place
in December 2004, which is typically the end of the BB season over the SWIO. Due to the long lifetime of $CH_3CN$ and to a





lesser extent CH$_3$COCH$_3$, part of these concentrations in zone I may be originating from accumulation of BB plumes in the troposphere.

The low variability in EnRs, between different BB intrusions at RUN, for both CH$_3$CN and CH$_3$COCH$_3$ allows for charac-
terisation of mixing ratios in the marine boundary layer with small relative uncertainties (8.3% and 13.5% respectively). The local impact of $\Delta$CH$_3$CN in the SWIO MBL during the August BB episodes ($\sim$ 50 pptv) constitutes an increase of $\sim$ 60 – 150% over the SWIO as measured during the MANCHOT campaign (zone I: 80$\pm$20 pptv, zone III: 20$\pm$10 pptv). Acetone excesses are based on the assumption that acetone production in the BB plume is similar in the free troposphere and in the marine boundary layer. The excesses over the SWIO can reach up to 300 pptv, $\sim$ 30 – 75% above the backgrounds recorded
during MANCHOT.

The relatively short lifetimes of CH$_3$OH and C$_6$H$_6$ result in a larger variability of the enhancement ratios between different BB intrusions. This is reflected in the larger relative uncertainty in the calculated excesses over the SWIO (21.7% and 32.6% for CH$_3$OH and C$_6$H$_6$ respectively). Calculated $\Delta$CH$_3$OH over the SWIO are $\sim$ 0.5 ppbv, corresponding to an increase of 25% (zone I) to at least 100% (zone III) compared to the values recorded during MANCHOT (Colomb et al., 2009). The expected
$\Delta$C$_6$H$_6$ over the SWIO is 30 pptv. This is only a minor increase compared to zone I of the MANCHOT campaign (160$\pm$40 pptv) but constitutes a significant increase (150%) in zone III, further south over the SWIO.

Due to the short lifetime of carboxylic acids in the humid marine boundary layer, the method used above to estimate the BB impact on the SWIO is not valid for HCOOH CH$_3$COOH.

## 5   Conclusions

We have shown that BB plumes were recorded with the hs-PTR-MS instrument deployed at the high-altitude Maïdo observatory located in the South-West Indian Ocean. Six different episodes of biomass burning plumes have been identified and studied in August 2018 and August 2019. Enhancement ratios relative to CO have been calculated for CH$_3$CN (1.61 – 2.06 pptv ppbv$^{-1}$), HCOOH (17.5 – 33.8 pptv ppbv$^{-1}$), CH$_3$COCH$_3$ (6.84 – 10.0 pptv ppbv$^{-1}$), CH$_3$COOH (9.8 – 18.0 pptv ppbv$^{-1}$), C$_6$H$_6$ (0.27
– 0.83 pptv ppbv$^{-1}$), CH$_3$OH (8.7 – 18.8 pptv ppbv$^{-1}$) and O$_3$ (410 – 640 pptv ppbv$^{-1}$). Comparison between these EnRs and the ERs calculated from literature showed production of CH$_3$COCH$_3$ and HCOOH. Secondary production of CH$_3$COCH$_3$ was accounted for by pyrogenic emission of precursor species propane and to a lesser extent i-butane and i-butene. Production was especially significant for HCOOH with EnRs about 10 times larger than the ERs. This HCOOH production can not be accounted for by known precursor species.
The CAMS NRT atmospheric composition service was shown to reproduce well the CO concentrations at RUN both during and outside BB episodes. In contrast, O$_3$ concentrations were only correctly reproduced outside the BB episodes. The large underestimation of O$_3$ concentrations during the BB episodes were linked to i) large uncertainties in VOC and NO$_x$ emissions and ii) misrepresentation of NO$_x$ recycling during transport of the BB plume in the CAMS NRT service. FLEXPART-AROME mesoscale backtrajectory simulations showed that biomass burning plumes were diluted at the observatory when the impact of

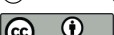



PBL air increased. This implies that the BB plume recorded at the observatory is primarily transported through the FT. Large scale transport of $\Delta CO$ originating from pyrogenic emissions, simulated with FLEXPART supported this by showing larger $\Delta CO$ concentrations at higher altitudes. Finally, the horizontal distribution of $\Delta CO$ in the SWIO MBL — calculated with FLEXPART — is multiplied with the EnR values inferred from data at the Maïdo observatory. This provided estimates for the impact of BB on air mass composition in the MBL over the SWIO. We compared the calculated estimates with background

VOC measurements in the region reported in literature. Expected excesses for $CH_3CN$, $CH_3COCH_3$, $H_6H_6$ and $CH_3OH$ represent an increase of background concentrations by $60 - 150\%$, $30 - 75\%$, $15 - 150\%$ and $25 - >100\%$, respectively. In the future, synchronous VOC measurements at RUN and marine campaigns should be conducted in order to i) better quantify the Ocean–Atmosphere interaction in regions with locally enhanced atmospheric concentrations of these species from BB and ii) identify the different ageing mechanisms during transport in the MBL compared to transport in the FT. This would be especially

valuable for $CH_3COCH_3$ and $CH_3OH$, for which the role of the ocean on the total atmospheric budget remains uncertain.

*Data availability.* The core hs-PTR-MS dataset is available at https://octave.aeronomie.be/index.php/datasets (last access August 2020). Other data is available upon request.

*Author contributions.* Formal analysis, B.V.; hs-PTR-MS data acquisition and management, C.A., N.S.; Thermo Scientific model 49i data acquisition and management, A.C., J.-M.M.; Picarro G2401 data acquisition and management, N.C., J.-M.M.; CU MAX-DOAS data aquisi-

tion and management, R.V., T.K.K. and C.F.L.; FLEXPART and FLEXPART-AROME simulations, B.V. and J.B.; original draft preparation, B.V.; review and editing, C.A., J.-F.M., T.S., J.B., R.V., N.S. and A.C.

*Competing interests.* The authors declare that they have no real or perceived conflicts of interests.

*Acknowledgements.* This research has been supported by the "Belgian Research Action through Interdisciplinary Networks" (BRAIN-be) through the Belgian Science Policy Office (BELSPO) (grant no. BR/175/A2/OCTAVE). The deployment of the PTR-MS at Maïdo is part of

a project that has received funding from the European Union's Horizon 2020 research and innovation program under grant agreement No. 654109. We would like to thank UMS3365 of OSU-Réunion for its support to the deployment of the hs-PTR-MS at Maïdo. R.V. acknowledges financial support for U.S. National Science Foundation award AGS-1620530. This work contains modified Copernicus Atmosphere Monitoring Service Information, neither the European Commission nor ECMWF is responsible for any use that may be made of the information it contains.



## Appendix A: In situ measurement data visualisation

### A1 Seasonal biomass burning profile

Hourly averages of CO and $CH_3CN$ are shown in Fig. A1. Both CO and $CH_3CN$ have large peak values from August to November. This corresponds to the biomass burning season as determined from ground-based remote-sensing data studies performed at La Réunion (Duflot et al., 2010; Vigouroux et al., 2012). The analysis presented in this study focuses on the first biomass burning intrusions measured for each season. The motivation for this choice is that the variability in diel profiles between different days is less pronounced during this period and backgrounds do not suffer from accumulated BB tracers for compounds with long atmospheric lifetimes.

### A2 Austral winter variation of in situ measurements at RUN

The temporal evolution of biomass burning plume constituents during austral winter 2018 and austral winter 2019 are shown together with the diel distribution of hourly averaged mixing ratio from Fig. A2 to Fig. A10. The median diel profile is used as an estimate of background variation above which the biomass burning excesses are determined. This works especially well for compounds with relatively small variability between different days compared to the excesses due to biomass burning (e.g. CO, $CH_3CN$, HCOOH, $CH_3COCH_3$ and $CH_3COOH$) but may introduce errors for other compounds (e.g. $C_6H_6$, $CH_3OH$ and $O_3$). When this difference becomes negligible, the analysis no longer works and these compounds are not considered (e.g. HCHO and $CH_3CHO$).

## Appendix B: $NO_2$ coincidence with FLEXPART simulations

$NO_2$ volume mixing ratios from the CU MAX-DOAS instrument are generally lower than 100 pptv outside of the BB episodes. A notable exception to this is 3 August 2019 when it reaches $\sim$ 280 pptv. This coincides with slightly elevated $\Delta CO$ signals simulated by FLEXPART at RUN (Fig. A11). At the visible wavelengths, the horizontal spatial scale probed is about 40 km and the overlap with the PBL is only a few km. As a results, measurements from the CU MAX-DOAS instrument are expected to compare well to the FLEXPART and CAMS models which have a low spatial resolution. Remark that when the $NO_2$ mixing ratio from the CU MAX-DOAS instrument is above 100 pptv, FLEXPART $\Delta CO$ is generally enhanced between 1000 – 1500 m a.g.l.

As the plume on 3 August is not clearly observed in the in situ measurements we assume that it is not well mixed with boundary layer air at RUN and do not investigate it further here.



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





**Table 1.** In situ routine observations at RUN and their respective limits of detection (LoD) and statistical uncertainties ($\sigma$). (Zhou et al., 2018; Duflot et al., 2019).

| Compound | Instrument | Technique | LoD [ppbv] | $\sigma$ [ppbv] |
|---|---|---|---|---|
| CO | Picarro G2401 | Cavity ring down spectroscopy | 1 | 1.5 |
| $O_3$ | Thermo Scientific model 49i | UV photometric analyser | 0.05 | 1 |





**Table 2.** A list of mass-to-charge ratios (m/z) observed in multiple ion detection mode by the hs-PTR-MS at RUN with the associated chemical compounds, dwell times and information about the instrument calibration limit of detection (LoD) per compound (Y: yes, N: no, N/A: not applicable). Dwell time is shown for 1 cycle. The LoD is computed for the hourly averages used here. The corresponding accumulated dwell times are about 22 time the stated dwell times of an individual cycle.

| m/z | Compound | Dwell time [s] | Calibrated | LoD [pptv] |
|---|---|---|---|---|
| 21 | $H_3{}^{18}O^+$ | 2 | N/A | N/A |
| 31 | formaldehyde (HCHO) | 10 | Y | 100 |
| 32 | $O_2^+$ | 0.1 | N/A | N/A |
| 33 | methanol ($CH_3OH$) | 10 | Y | 40 |
| 37 | $H_3O^+.H_2O$ | 0.1 | N/A | N/A |
| 42 | acetonitrile ($CH_3CN$) | 10 | Y | 1 |
| 45 | acetaldehyde ($CH_3CHO$) | 10 | Y | 18 |
| 47 | formic acid (HCOOH) | 10 | N | 50 |
| 59 | acetone ($CH_3COCH_3$) | 10 | Y | 4 |
| 61 | acetic acid ($CH_3COOH$) | 10 | N | 7 |
| 63 | dimethyl sulphide (DMS) | 10 | Y | 6 |
| 69 | isoprene ($C_5H_8$) | 10 | Y | 5 |
| 71 | methyl vinyl ketone (MVK)/ methacrolein (MACR)/ hydroxy hydroperoxides from isoprene (ISOPOOH) | 10 | Y | 2 |
| 73 | methyl ethyl ketone (MEK) | 10 | Y | 3 |
| 79 | benzene ($C_6H_6$) | 10 | Y | 2 |
| 81 | sum of monoterpenes[α] ($C_{10}H_{16}$) | 10 | Y | 5 |
| 93 | toluene ($C_7H_8$) | 10 | Y | 7 |
| 107 | xylenes[α] ($C_8H_{10}$) | 10 | Y | 7 |
| 137 | sum of monoterpenes[α] ($C_{10}H_{16}$) | 10 | Y | 8 |

[α]o-xylene and limonene were used for calibration.





**Table 3.** Pearson correlation coefficients ($r$) between the excess of chemical compound X ($\Delta$X) and the excess of the typical BB marker $CH_3CN$ during the BB episodes.

| X | $r$ | X | $r$ |
|---|---|---|---|
| CO | 0.98 | DMS | 0.60 |
| HCOOH | 0.89 | HCHO | 0.55 |
| $CH_3COCH_3$ | 0.88 | MEK | 0.39 |
| $CH_3COOH$ | 0.87 | $CH_3CHO$ | 0.12 |
| $O_3$ | 0.83 | $C_5H_8$ | -0.08 |
| $C_6H_6$ | 0.81 | MVK/MACR/ISOPOOH | -0.22 |
| $CH_3OH$ | 0.71 | | |

**Table 4.** Mean background mixing ratio [ppbv] during the daytime (10 a.m. − 4 p.m., reflecting planetary boundary layer air composition) and nighttime (10 p.m. − 4 a.m., measuring free tropospheric air masses), $\mu_{PBL}$ and $\mu_{FT}$ respectively, and mean excesses [%] during the Aug 2018 and Aug 2019 BB intrusions. The reported background values are mean mixing ratios recorded during austral winter (June, July, August), excluding BB incidents.

| | $\mu_{PBL}$ [ppbv] | $\mu_{FT}$ [ppbv] | 03-05 Aug 2018 | 08-14 Aug 2018 | 17-19 Aug 2018 | 06-08 Aug 2019 | 10-11 Aug 2019 | 15-18 Aug 2019 |
|---|---|---|---|---|---|---|---|---|
| $CH_3CN$ | 0.091 (0.035) | 0.092 (0.042) | 96% | 164% | 99% | 182% | 128% | 103% |
| CO | 67 (15) | 61 (17) | 81% | 129% | 78% | 140% | 110% | 68% |
| HCOOH | 0.87 (0.62) | 0.40 (0.62) | 466% | 630% | 285% | 942% | 515% | 379% |
| $CH_3COCH_3$ | 0.40 (0.17) | 0.30 (0.19) | 153% | 221% | 118% | 227% | 157% | 147% |
| $CH_3COOH$ | 0.40 (0.30) | 0.17 (0.31) | 709% | 893% | 398% | 1155% | 754% | 450% |
| $C_6H_6$ | 0.025 (0.014) | 0.012 (0.012) | 303% | 314% | 123% | 360% | 274% | 156% |
| $CH_3OH$ | 1.2 (0.42) | 0.63 (0.36) | 131% | 163% | 63% | 188% | 132% | 101% |



**Table 5.** Calculated EnRs [pptv ppbv$^{-1}$] — relative to CO — for the identified BB intrusions. Between brackets are the standard deviations of the EnR [pptv ppbv$^{-1}$] obtained from the linear fit.

| | 3-5 Aug 2018 | 8-14 Aug 2018 | 17-19 Aug 2018 | 6-8 Aug 2019 | 10-11 Aug 2019 | 15-18 Aug 2019 |
|---|---|---|---|---|---|---|
| $CH_3CN$ | 1.61 (0.02) | 1.71 (0.01) | 1.69 (0.04) | 1.79 (0.01) | 1.69 (0.04) | 2.06 (0.03) |
| HCOOH | 29.3 (0.5) | 23.5 (0.3) | 17.5 (0.5) | 33.8 (0.5) | 24.6 (0.9) | 31.2 (0.6) |
| $CH_3COCH_3$ | 8.85 (0.22) | 7.76 (0.13) | 6.84 (0.14) | 7.85 (0.08) | 7.06 (0.26) | 10.0 (0.3) |
| $CH_3COOH$ | 18.0 (0.3) | 12.9 (0.2) | 9.8 (0.3) | 16.3 (0.2) | 13.2 (0.5) | 13.7 (0.2) |
| $O_3$ | 640 (19) | 438 (9) | 635 (27) | 461 (9) | 410 (20) | 422 (16) |
| $C_6H_6$ | 0.83 (0.04) | 0.46 (0.01) | 0.27 (0.01) | 0.50 (0.01) | 0.50 (0.02) | 0.46 (0.02) |
| $CH_3OH$ | 18.8 (0.6) | 13.5 (0.4) | 8.7 (0.5) | 15.9 (0.3) | 14.2 (0.8) | 16.8 (0.7) |



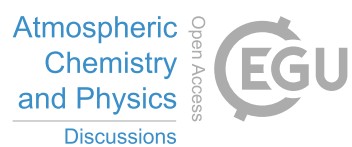

**Table 6.** ER [pptv ppbv$^{-1}$] — relative to CO — for the 3 most probable fuel types (Savanna and grassland, tropical forest and agricultural residue) of the BB plume sampled at RUN. Between brackets is the uncertainty obtained by combining the standard deviations of the EFs recorded by Andreae (2019).

| | $CH_3CN$ | HCOOH | $CH_3COOH$ | $CH_3COCH_3$ | $C_6H_6$ | $CH_3OH$ |
|---|---|---|---|---|---|---|
| Savanna and grassland | 1.68 (1.62) | 1.85 (1.59) | 15.6 (10.3) | 3.28 (2.24) | 1.72 (0.87) | 17.1 (21.1) |
| Tropical forest | 3.21 (3.10) | 2.87 (2.47) | 14.8 (9.76) | 2.92 (-) | 1.31 (0.664) | 23.5 (29.1) |
| Agricultural residue | 2.24 (2.17) | 4.48 (3.86) | 37.4 (24.7) | 4.5 (3.07) | 1.27 (0.646) | 38.0 (46.9) |
| Vigouroux et al. (2012)[α] | | 4.6 (0.3) | | | | 11.6 (0.6) |
| de Gouw et al. (2006)[β] | 1.18 (0.14) – 3.24 (0.09) | | 0.9 (0.3) – 12.9 (0.5) | 2.6 (0.3) – 22.8 (1.0) | 0.8 (0.2) – 1.41 (0.04) | 2 (2) – 21 (2) |
| Lefer et al. (1994)[γ] | | 8.2 – 62 | 6 – 34 | | | |
| This work[δ] | 1.76 (0.146) | 26.7 (5.44) | 14.0 (2.61) | 8.06 (1.09) | 0.50 (0.16) | 14.6 (3.18) |

[α] EnRs derived from FTIR measurements at RUN.
[β] EnR ranges from 11 forest fire plumes Sampled during NEAQS-ITCT 2k4.
[γ] EnR ranges from 10 subarctic forest fire plumes sampled during ABLE 3B.
[δ] This work, mean EnR and the standard deviation.

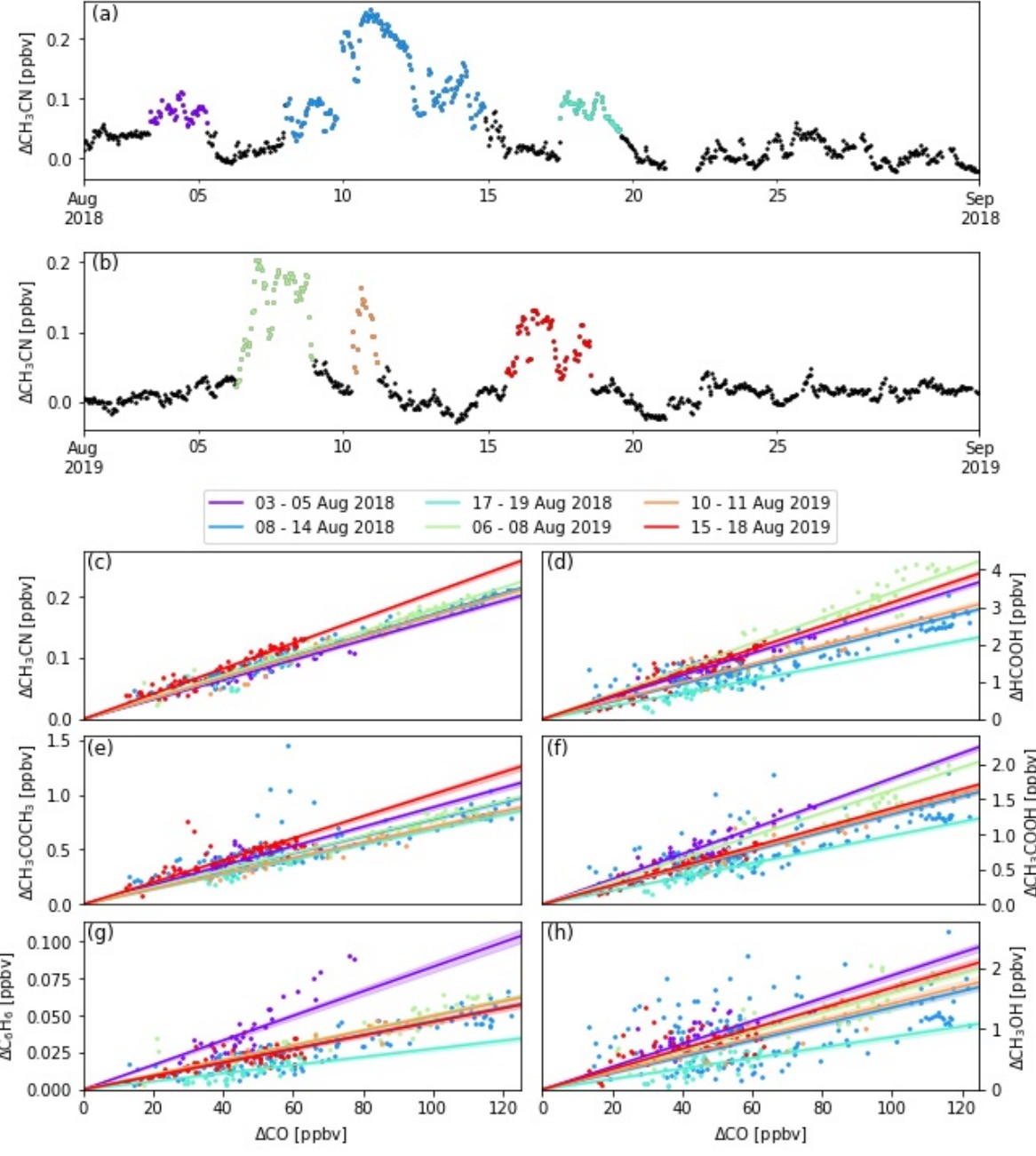

**Figure 1.** The top two panels (a and b) show the six BB intrusions identified using $\Delta CH_3CN$ (purple: 03 – 05 Aug 2018, blue: 8 – 14 Aug 2018, cyan: 17 – 19 Aug 2018, green: 6 – 8 Aug 2019, orange: 10 – 11 Aug 2019, red: 15 – 18 Aug 2019). The bottom six panels show the EnR fits for $CH_3CN$ (c), $HCOOH$ (d), $CH_3COOH$ (e), $CH_3COCH_3$ (f), $C_6H_6$ (g) and $CH_3OH$ (h). EnRs are normalised to the excess mixing ratio of CO for the six intrusions in the colors used in the top two panels. Uncertainty on the linear regression is shown as a coloured band around the curves.



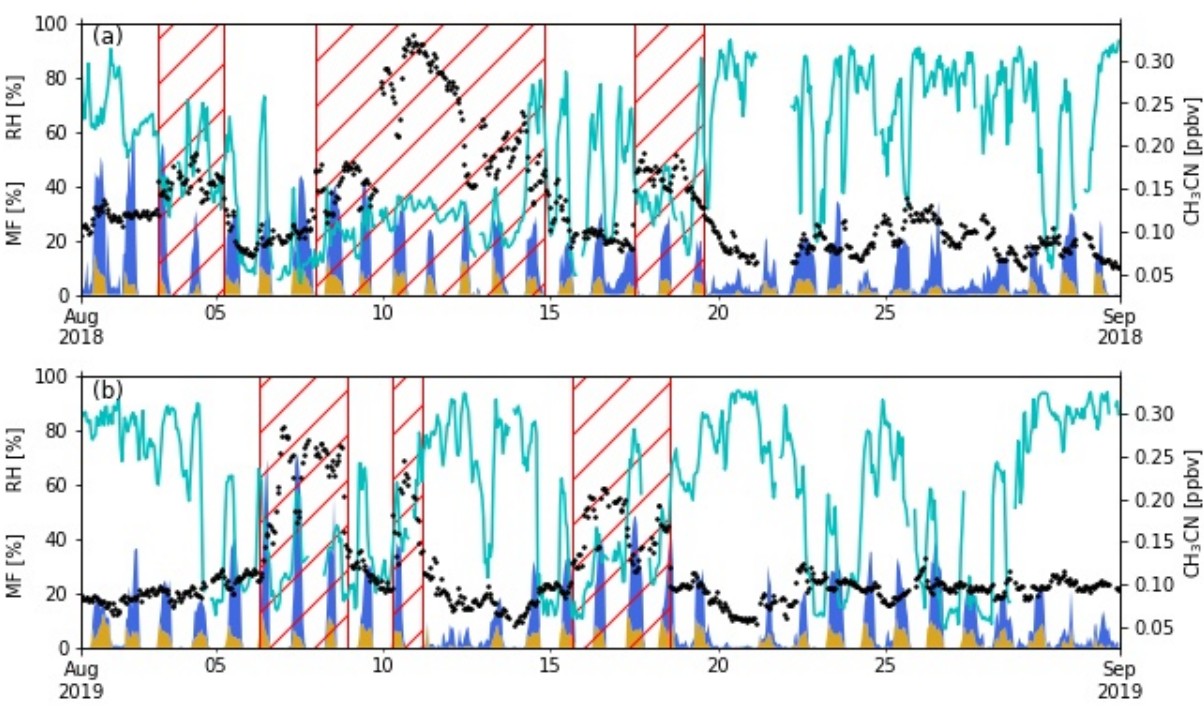

**Figure 2.** Measured relative humidity (RH, cyan curve), CH$_3$CN mixing ratio [ppbv] (black points) and modelled mesoscale MF [%] from 24 hour backtrajectories using FLEXPART-AROME in August 2018 (a) and August 2019 (b). Blue denotes the marine boundary layer MF, brown represents the island surface PBL MF. The hatched red area represents the different BB intrusions.

**Figure 3.** Comparison between measured (black dots) and calculated mixing ratios (coloured lines) of CO (a), $O_3$ (b) and $NO_2$ (c) from CAMS. The coloured lines indicate the lowest eight pressure levels in the model (1000, 950, 925, 900, 850, 800, 700 and 600 mbar). The hatched red area represents the different BB intrusions.

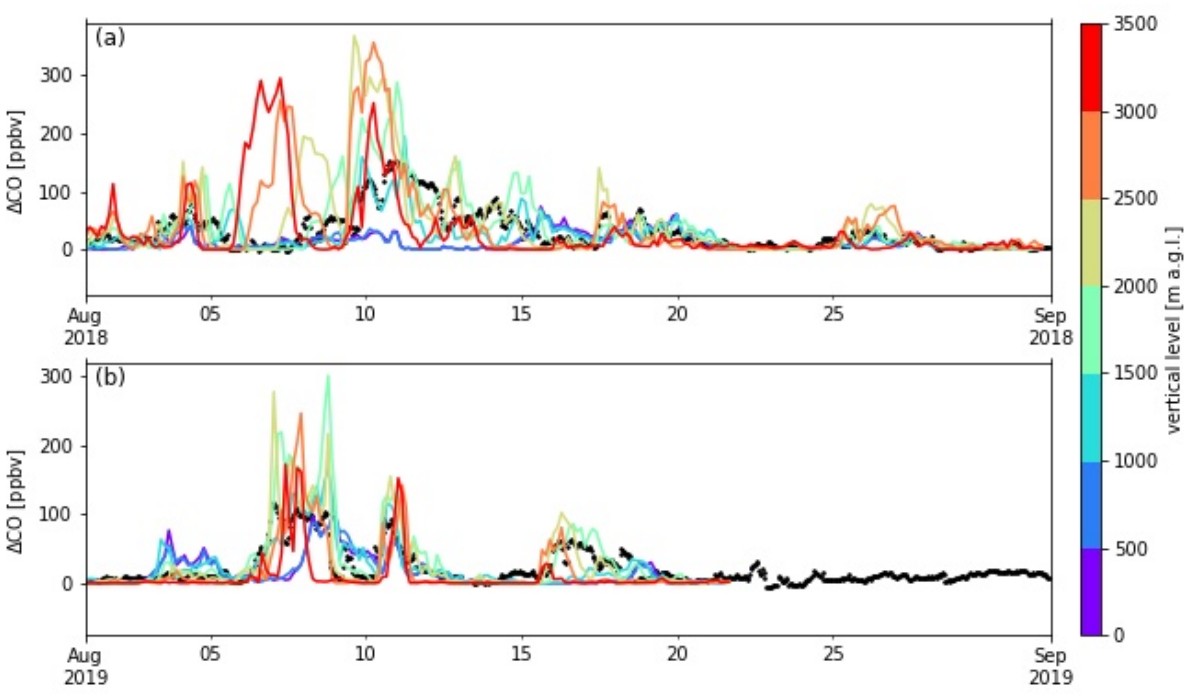

**Figure 4.** Comparison between measured excess CO (black) at RUN to that modelled on different vertical levels of the FLEXPART model during August 2018 (a) and August 2019 (b). FLEXPART output levels are defined in meters above ground level (m a.g.l.).



**Figure 5.** Excess CO over the South West Indian Ocean between 0 and 500 m a.g.l. from BB emissions as simulated by FLEXPART. Additional color scales quantify the projected $CH_3CN$, $CH_3COCH_3$, $C_6H_6$ and $CH_3OH$ excesses.





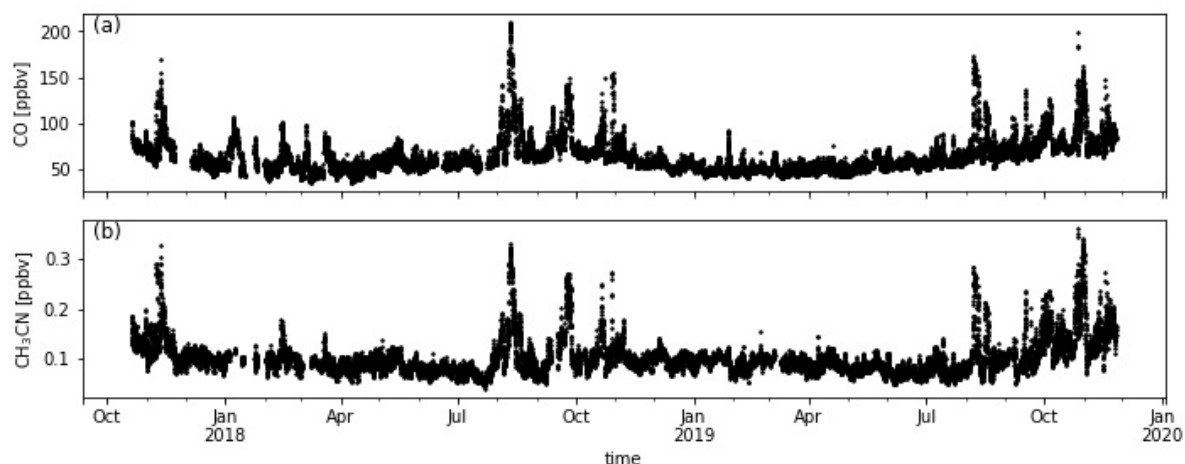

**Figure A1.** Measured CO (a) and CH$_3$CN (b) mixing ratios [ppbv] during the deployment of the hs-PTR-MS for the OCTAVE project.





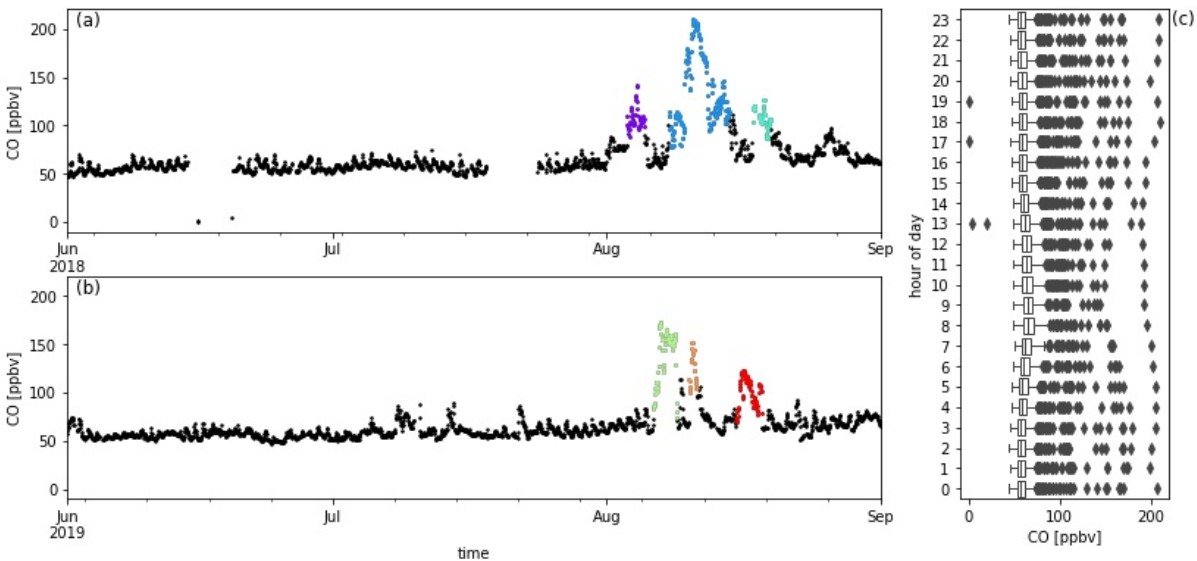

**Figure A2.** Measured CO mixing ratios [ppbv] during austral winter 2018 (a) and austral winter 2019 (b) together with the diel distribution of hourly averages (c). Biomass burning plumes under investigation are highlighted in colors.



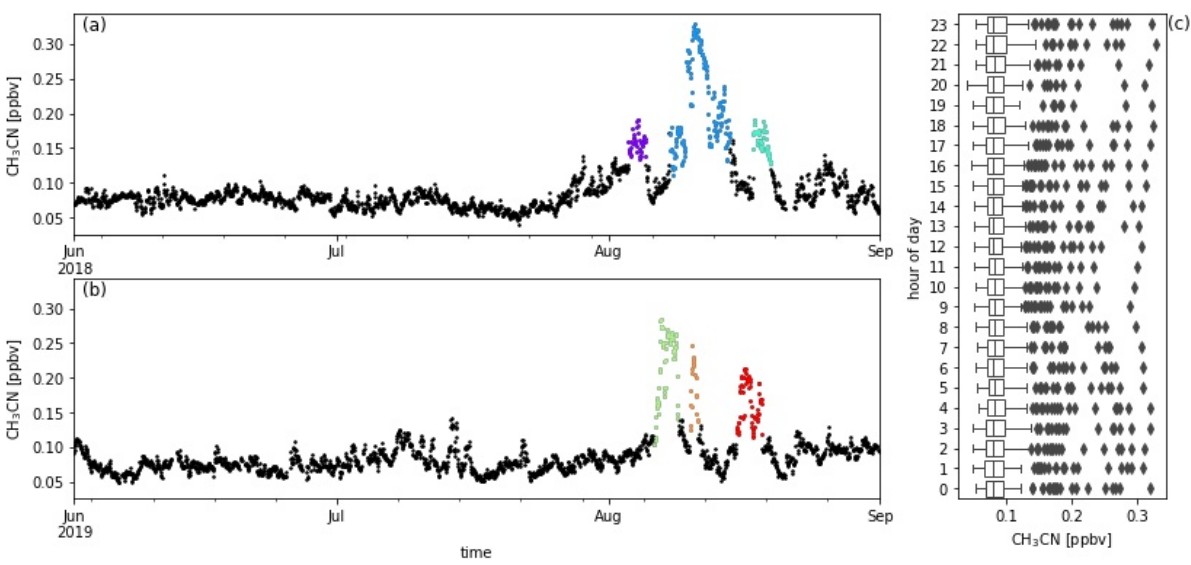

**Figure A3.** Measured CH$_3$CN mixing ratios [ppbv] during austral winter 2018 (a) and austral winter 2019 (b) together with the diel distribution of hourly averages (c). Biomass burning plumes under investigation are highlighted in colors.





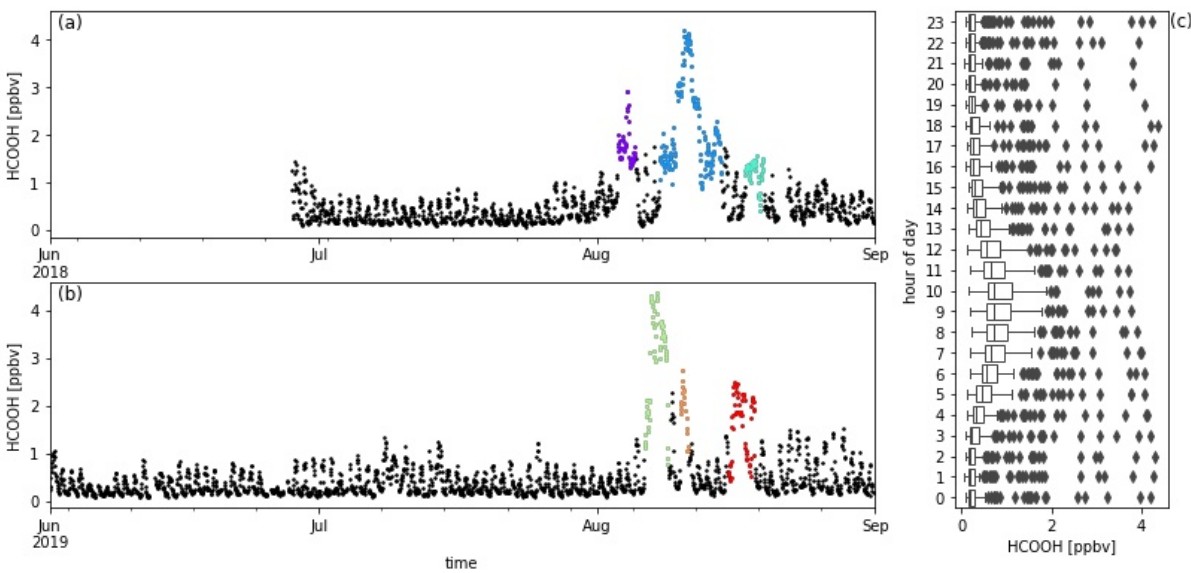

**Figure A4.** Measured HCOOH mixing ratios [ppbv] during austral winter 2018 (a) and austral winter 2019 (b) together with the diel distribution of hourly averages (c). Biomass burning plumes under investigation are highlighted in colors.



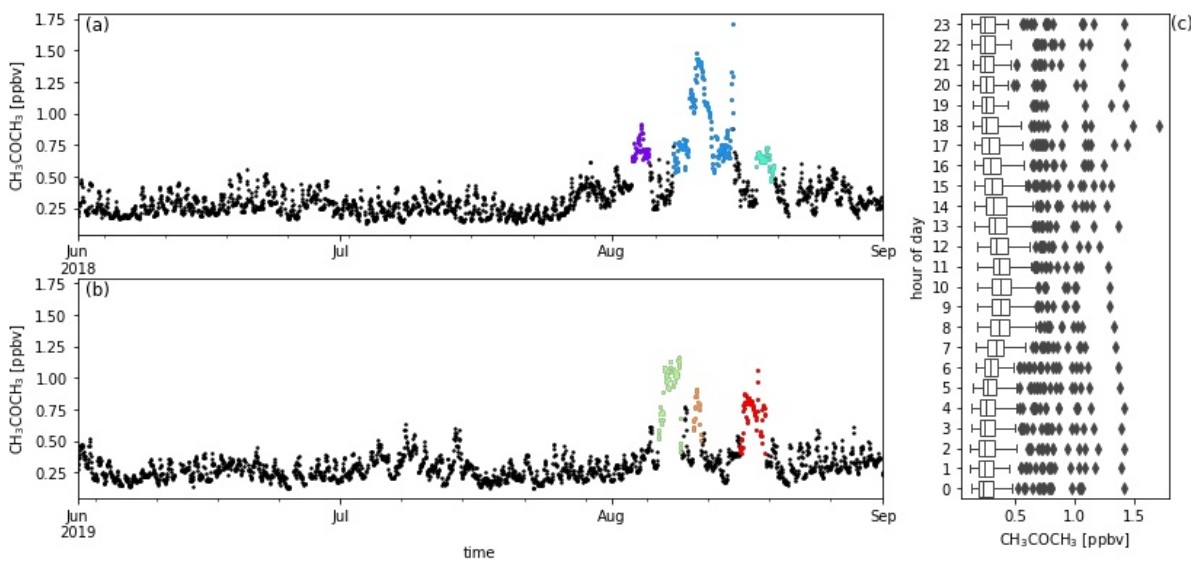

**Figure A5.** Measured $CH_3COCH_3$ mixing ratios [ppbv] during austral winter 2018 (a) and austral winter 2019 (b) together with the diel

distribution of hourly averages (c). Biomass burning plumes under investigation are highlighted in colors.



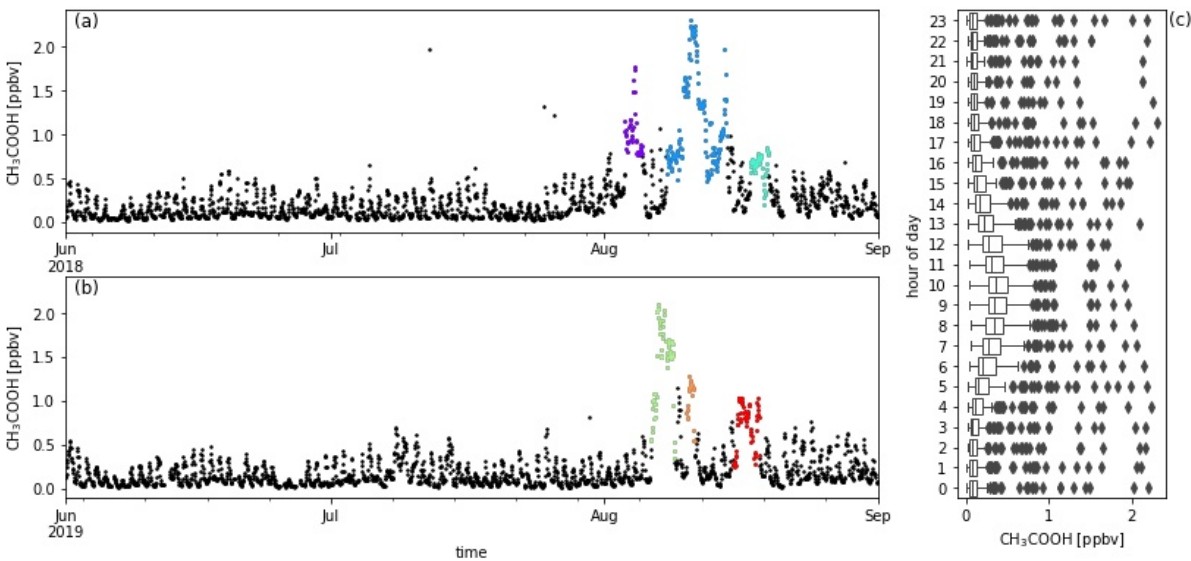

**Figure A6.** Measured CH$_3$COOH mixing ratios [ppbv] during austral winter 2018 (a) and austral winter 2019 (b) together with the diel distribution of hourly averages (c). Biomass burning plumes under investigation are highlighted in colors.





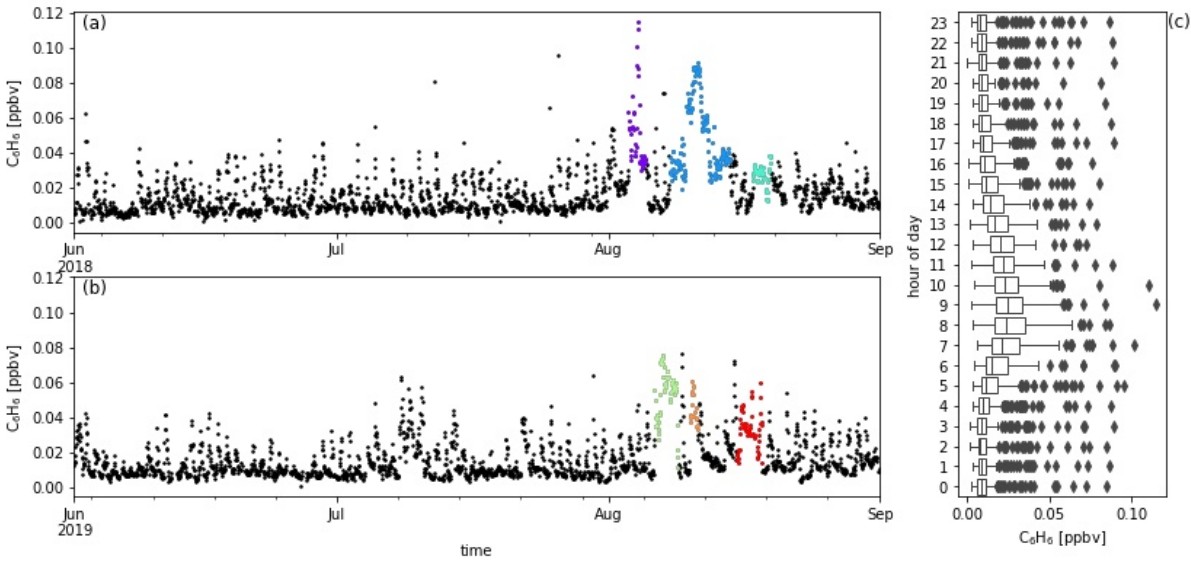

**Figure A7.** Measured $C_6H_6$ mixing ratios [ppbv] during austral winter 2018 (a) and austral winter 2019 (b) together with the diel distribution of hourly averages (c). Biomass burning plumes under investigation are highlighted in colors.

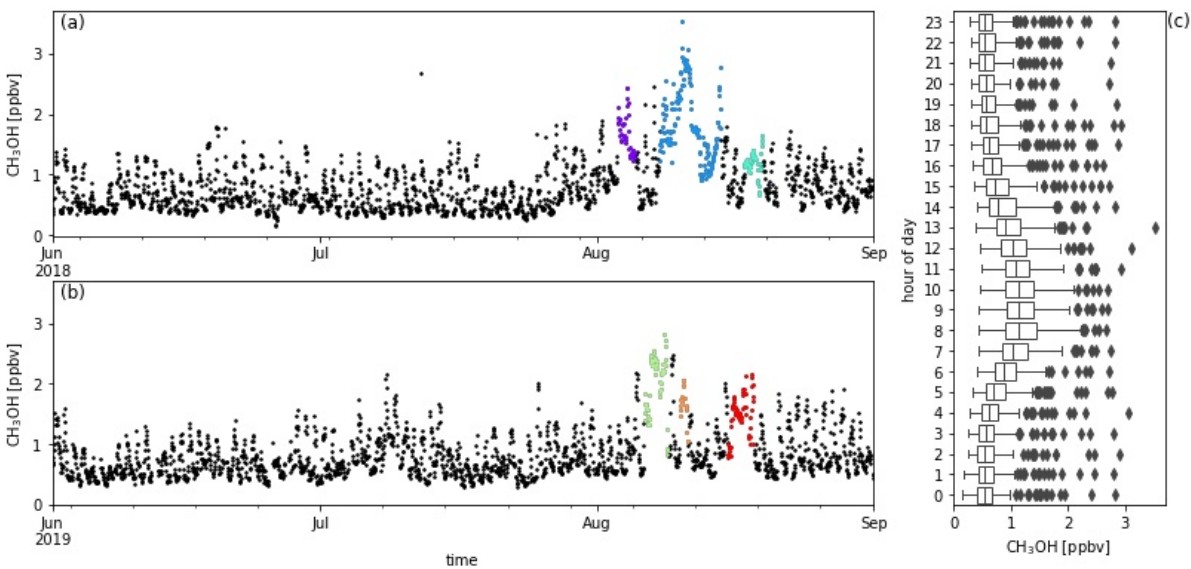

**Figure A8.** Measured CH$_3$OH mixing ratios [ppbv] during austral winter 2018 (a) and austral winter 2019 (b) together with the diel distribution of hourly averages (c). Biomass burning plumes under investigation are highlighted in colors.




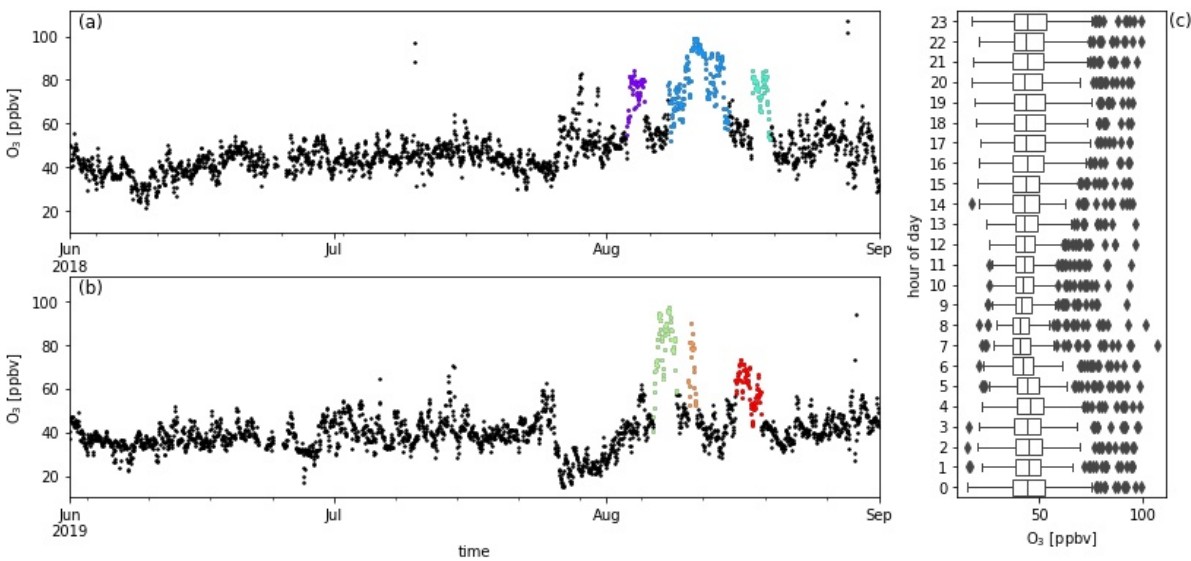

**Figure A9.** Measured $O_3$ mixing ratios [ppbv] during austral winter 2018 (a) and austral winter 2019 (b) together with the diel distribution of hourly averages (c). Biomass burning plumes under investigation are highlighted in colors.





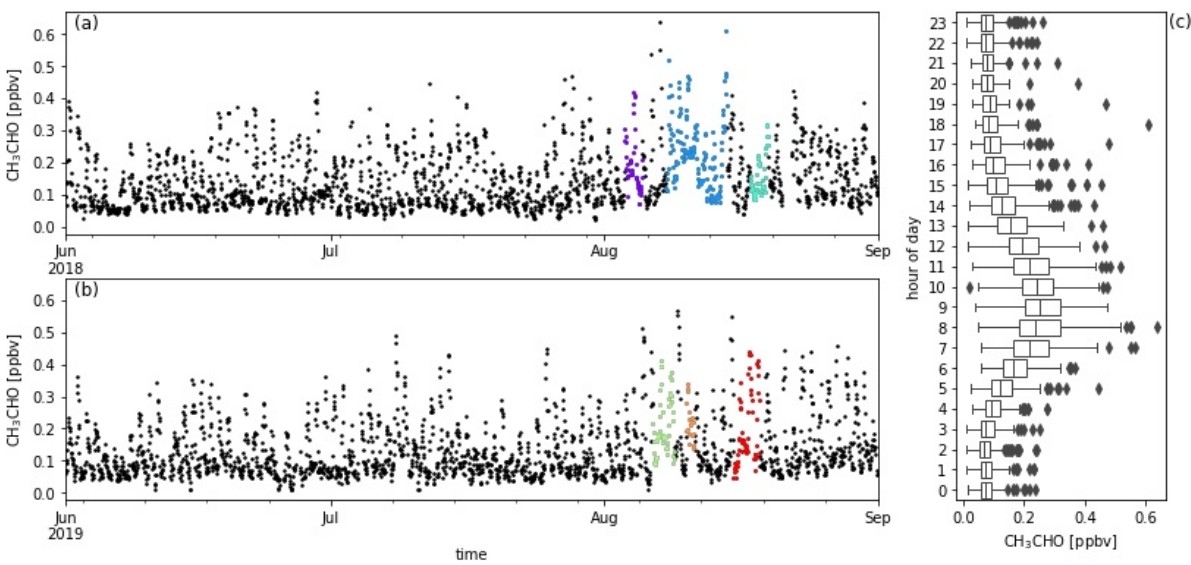

**Figure A10.** Measured CH$_3$CHO mixing ratios [ppbv] during austral winter 2018 (a) and austral winter 2019 (b) together with the diel distribution of hourly averages (c). Biomass burning plumes under investigation are highlighted in colors.



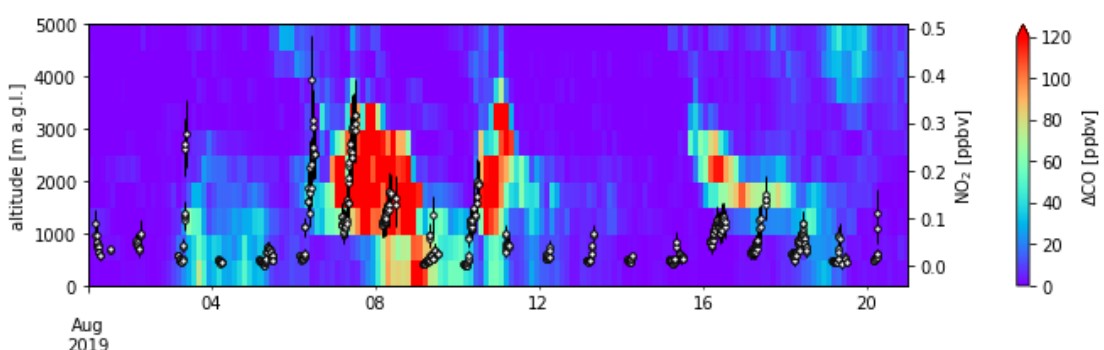

**Figure A11.** NO$_2$ [ppbv] from CU MAX-DOAS in comparison with $\Delta$CO [ppbv] from FLEXPART simulations. Altitude [m a.g.l.] corresponds with vertical layers in the FLEXPART output at the location of RUN. The ground level of RUN in the model is 284 m a.s.l.