# Peer review of "Characterisation of African biomass burning plumes and impacts on the atmospheric composition over the South-West Indian Ocean"

_Atmospheric Chemistry and Physics, 2020_

## Referee Comment (RC1) · Anonymous Referee #1 · 26 Aug 2020

General remarks: This is a well-written sound paper describing a campaign in Maïdo for testing the effect of bio-mass burning on mixing ratios of VOCs and on the model performance if a chemical transport model to calculate levels of O3 and NO2 at the site. By comparing the ratio of measured VOCs against CO and emission ratios, assessments about losses and production during transport are made. In general, the conclusions are a little bit weak but I am not aware of any similar data from this region of the world and therefore, I suggest publishing the manuscript in ACP, taking into account the suggestions from below.

Major Issue: L9, For C6H6 and CH3OH, the EnR is lower than the ER, indicating a

[Figure]

significant net sinks of these compounds. This is overselling as you assess it in line 274ff. as completely in line with the lifetimes and the transport time. I was completely mislead when I read it in the abstract only. I thought you found an exceptional sink but I was then disappointed when I came to line 274.

Minor issues: Abstract: This is obviously a paper focusing on VOC. Thus, mention the VOCs first.

L35: there is also incomplete understanding about direct emissions.

L37: Isn't CO also controlling the O3 levels in remote atmospheres?

L65 ff: only OVOCs and certain VOCs with a conjugated double bond or heteroatoms can be measured by PTR-MS. Mention somewhere that other VOCs could also be present in BB plumes but were not in the focus of this study

L249: the primary sink of what?

L254: . . .CH3OH and are shown. . .

L296ff: a part of the difference could be that you did not measure the NMHCs, which also have BB sources and could contribute to the O3 formation. This could also be mentioned again in the conclusion, e.g. L 367.

Table 4: As in the legend for Table 5, mention the values in the brackets.

Figure 2: explain MF

Figure 5: Please indicate Maïdo on the map for those being weak in geography of the South West Indian Ocean

[Figure]

---

## Referee Comment (RC2) · Anonymous Referee #2 · 22 Sep 2020

General comments:

The manuscript is well written with sound scientific results and an abundance of figures and tables to support the text.

I suggest it is published to ACP, after addressing these minor technical details.

Technical comments: Figure 1: The uncertainty is barely visible in the bottom 6 plots (c,d,e,f,g,h). In most cases it looks as part of the pixelation (maybe increase resolution?)

Table 5 and Table 6 caption: You mention brackets for the standard deviation, when in

the table you use parentheses.

---

## Author Comment (AC1) · 6 Oct 2020

We would like to thank the referees for their reviews. The manuscript has been revised according to their comments.

When rereading the manuscript we noticed that the altitude of the Maïdo observatory has been consistently reported as 2250 a.s.l. This is an error, the real altitude of the observatory is 2160 m a.s.l. This has been corrected everywhere in the manuscript and does not affect the analysis or the discussion presented in the current work.

**Anonymous Referee #1**

This is a well-written sound paper describing a campaign in Maïdo for testing the effect of biomass burning on mixing ratios of VOCs and on the model performance if a chemical transport model to calculate levels of O3 and NO2 at the site. By comparing the ratio of measured VOCs against CO and emission ratios, assessments about losses and production during transport are made. In general, the conclusions are a little bit weak but I am not aware of any similar data from this region of the world and therefore, I suggest publishing the manuscript in ACP, taking into account the suggestions from below.

Major issue:

*L9:* For C6H6 and CH3OH, the EnR is lower than the ER, indicating a significant net sinks of these compounds. This is overselling as you assess it in line 274ff. as completely in line with the lifetimes and the transport time. I was completely mislead when I read it in the abstract only. I thought you found an exceptional sink but I was then disappointed when I came to line 274.
We have nuanced this in the abstract and explicitly mention now that this is in line with the expected atmospheric lifetime. Thank you for this reaction, we did not want to mislead any readers.

Minor issues:

*Abstract*: This is obviously a paper focusing on VOC. Thus, mention the VOCs first.
The order has been changed.

*L35*: there is also incomplete understanding about direct emissions.
The text has been completed.

*L37*: Isn't CO also controlling the O3 levels in remote atmospheres?
We do not expect CO to control the ozone levels in the remote atmosphere as you need NOx for ozone production during the OH initiated CO oxidation. However, the mixing ratios of CO, as well as CH4, have a major impact on the oxidative capacity of the remote marine atmosphere as they account for over 50% of the OH reactivity [Travis et al, 2020]. We refer here specifically to the role of OVOCs compared to all other NMVOCs. This specification has been added to the article.

*L65ff*: only OVOCs and certain VOCs with a conjugated double bond or heteroatoms can be measured by PTR-MS. Mention somewhere that other VOCs could also be present in BB plumes but were not in the focus of this study.
We have added that other compounds could have been present in biomass burning plumes but that either they were not the focus of this study or local sources interfered too much in order for us to reliably quantify the enhancement ratios (e.g. MEK).

*L249*: the primary sink of what?
Of HCOOH when the wet deposition sink is negligible in the free troposphere during the dry season. The information relating to a specific compound was taken out during a rewrite to generalise the discussion in this section. Thank you for noticing this. We have put the focus on the reduction of the wet deposition sink which is a statement that holds up for all compounds.

*L254*: . . .CH3OH and are shown. . .

We have restructured the sentence to be grammatically correct.

*L296ff*: a part of the difference could be that you did not measure the NMHCs, which also have BB sources and could contribute to the O3 formation. This could also be mentioned again in the conclusion, e.g. L 367.
The corresponding line reference is discussing HCOOH formation during transport. Measuring a complete array of NMHCs could indeed help identify precursors present in the plume. However, note that when production of HCOOH is a fast process, the precursor might have been mostly/ fully lost during transport.
Instead of focussing on measurements, we searched for known HCOOH precursors in the literaⁱ ture and cross-referenced these with species emitted by biomass burning sources using the emission factors compiled by Adreae (2019). This is the same approach we used in looking for secondary production of CH3COCH3 during transport. We have added this clarification to the manuscript.

*Table 4*: As in the legend for Table 5, mention the values in the brackets.
It has been adjusted.

*Figure 2*: explain MF
Ok.

*Figure 5*: Please indicate Maïdo on the map for those being weak in geography of the South West Indian Ocean
The location of the Island has been added to the maps.

**Anonymous Referee #2**

The manuscript is well written with sound scientific results and an abundance of figures and taⁱ bles to support the text. I suggest it is published to ACP, after addressing these minor technical details.

Technical comments:
*Figure 1*: The uncertainty is barely visible in the bottom 6 plots (c,d,e,f,g,h). In most cases it looks as part of the pixelation (maybe increase resolution?)
We have regenerated the figures with a smaller linewidth and in pdf format to increase resolution. For consistency we have recreated all figures in pdf format, increasing the resolution for each.

*Table 5 and Table 6 caption*: You mention brackets for the standard deviation, when in the table you use parentheses.
Thank you for noticing, it has been corrected.

**References:**
* Travis, K. R., Heald, C. L., Allen, H. M., Apel, E. C., Arnold, S. R., Blake, D. R., Brune, W. H., Chen, X., Commane, R., Crounse, J. D., Daube, B. C., Diskin, G. S., Elkins, J. W., Evans, M. J., Hall, S. R., Hintsa, E. J., Hornbrook, R. S., Kasibhatla, P. S., Kim, M. J., Luo, G., McKain, K., Millet, D. B., Moore, F. L., Peischl, J., Ryerson, T. B., Sherwen, T., Thames, A. B., Ullmann, K., Wang, X., Wennberg, P. O., Wolfe, G. M., and Yu, F.: Constraining remote oxidation capacity with ATom observations, Atmos. Chem. Phys., 20, 7753–7781, https://doi.org/10.5194/acp-20-7753-2020, 2020.

* Andreae, M. O.: Emission of trace gases and aerosols from biomass burning – an updated asⁱ sessment, Atmos. Chem. Phys., 19, 8523–8546, https://doi.org/10.5194/acp-19-8523-2019, 2019.